# Dissociating task acquisition from expression during learning reveals latent knowledge

Kishore V. Kuchibhotla [1,2,9,10], Tom Hindmarsh Sten[3,4,8,9], Eleni S. Papadoyannis[3,4], Sarah Elnozahy[1], Kelly A. Fogelson[1], Rupesh Kumar[5], Yves Boubenec[5], Peter C. Holland[1,2], Srdjan Ostojic [6] & Robert C. Froemke [3,4,7,10]

Performance on cognitive tasks during learning is used to measure knowledge, yet it remains controversial since such testing is susceptible to contextual factors. To what extent does performance during learning depend on the testing context, rather than underlying knowledge? We trained mice, rats and ferrets on a range of tasks to examine how testing context impacts the acquisition of knowledge versus its expression. We interleaved reinforced trials with probe trials in which we omitted reinforcement. Across tasks, each animal species performed remarkably better in probe trials during learning and inter-animal variability was strikingly reduced. Reinforcement feedback is thus critical for learning-related behavioral improvements but, paradoxically masks the expression of underlying knowledge. We capture these results with a network model in which learning occurs during reinforced trials while context modulates only the read-out parameters. Probing learning by omitting reinforcement thus uncovers latent knowledge and identifies context- not "smartness"- as the major source of individual variability.

[1] Department of Psychological and Brain Sciences, Johns Hopkins University, Baltimore, MD 21218, USA. [2] Department of Neuroscience, Johns Hopkins Medical School, Baltimore, MD 21218, USA. [3] Departments of Otolaryngology, Neuroscience and Physiology, Skirball Institute, Neuroscience Institute, New York University School of Medicine, New York, NY 10016, USA. [4] Center for Neural Science, New York University, New York, NY 10003, USA. [5] Laboratoire des Systèmes Perceptifs, UMR8248, École Normale Supérieure-PSL Research University, 75006 Paris, France. [6] Laboratoire de Neurosciences Cognitives, INSERM U960, École Normale Supérieure-PSL Research University, 75006 Paris, France. [7] Faculty Scholar, Howard Hughes Medical Institute, Chevy Chase, MA 20815, USA. [8] Present address: Laboratory of Neurophysiology and Behavior, The Rockefeller University, New York, NY 10065, USA. [9] These authors contributed equally: Kishore V. Kuchibhotla, Tom Hindmarsh Sten. [10] These authors jointly supervised this work: Kishore V. Kuchibhotla, Robert C. Froemke. Correspondence and requests for materials should be addressed to K.V.K. (email: kkuchib1@jhu.edu)

Assessment of learning and aptitude often requires animals and humans to report their underlying knowledge at a given moment, often in specific testing environments or contexts[1]. In reinforcement learning paradigms, learning rates are inferred from behavioral reports[2]. Self-reporting, however, is highly sensitive to a variety of contextual factors unrelated to knowledge of the core task demands[3,4], potentially confounding the interpretation of behavioral performance. Animal models of learning have gained traction in recent years because they allow more direct links to be established between behavioral performance, computations and algorithms used for learning, and neural implementations of these algorithms[5]. The ability to monitor the activity of the same neurons over many days using chronic two-photon imaging[6] and single-unit electrophysiology[7] has further accelerated the exploration of neural mechanisms of learning. To date, studies focused on acquisition of task knowledge (i.e., learning rate) depend upon measuring expression of that knowledge by an animal's own self-report. This is true for sensorimotor tasks typically used across sensory modalities[6,8–13].

Moreover, a key tenant of behavioral and systems neuroscience posits that humans and other animals learn tasks at vastly different rates[6,10,14–17]. This has led to the idea that inter-animal variability in performance arises from differences in underlying learning rate parameters that impact the rate of task acquisition[17–19]. While attempts have been made to link learning-related performance variability to various modulatory factors[16,20], these approaches mostly focus on the acquisition of task-related contingencies.

Here, we dissociate the acquisition of underlying stimulus-action associations from context-dependent expression by manipulating the testing context. We introduce a simple behavioral manipulation, removing access to reinforcement (probe context), and then measure behavioral performance in two distinct contexts, one with reinforcement and the other without. Surprisingly, we identify two parallel learning processes, one related to the acquisition of task contingencies (revealed in the probe context) and the other related to the expression of that knowledge (demonstrated in the reinforced context). Across tasks, each animal species performed remarkably better in the probe context during learning and inter-animal variability was strikingly reduced. In the presence of reinforcement, performance trajectories were slower and far more variable between individual animals. As a result, we find that the probe context reveals a learning trajectory that more faithfully describes the acquisition of stimulus-action associations while the reinforced context demonstrates the role of contextual factors in modulating behavioral expression. We capture these results with a network model in which learning occurs during reinforced trials while context modulates only the read-out parameters. Reinforcement feedback is thus critical for learning-related plasticity but simultaneously can mask the expression of underlying knowledge.

## Results

### Expression of knowledge during learning is context-dependent.
To determine how context affects the behavioral assessment of learning, we first trained mice on an auditory go/no-go stimulus recognition task[21] (Fig. 1a). Mice learned to lick for a water reward provided through a lick tube after hearing a conditioned stimulus (the 'target' tone) and to withhold from licking after hearing an unrewarded ('foil') tone of a different frequency (Fig. 1b). Similar to a previous report[21], animals learned to perform the task at expert levels in the reinforced context over the course of multiple training sessions (Fig. 1c). At expert levels, mice consistently licked to the target tone (Fig. 1d) and withheld from licking to the foil tone (Fig. 1d, Supplementary Movie 1).

Over the course of learning, we interleaved the reinforced context with a smaller number of trials without reinforcement by removing the licktube ('probe context', Fig. 1e). In the probe context, we removed the licktube for a subset of trials (<40) in order to test whether absence of reinforcement would change the self-report of the mice. First, we focused on a trial block early in learning (trial block 1500–2000) when animals were tone responsive; i.e., they licked indiscriminately to both target and foil tones in the reinforced context, but did not lick during the inter-trial interval (Fig. 1f, 'reinforced context', Fig. 1g, Supplementary Movie 2; hits: $96.0 \pm 1.4\%$, false-alarms: $81.0 \pm 4.6\%$). Surprisingly, when we removed the licktube for the probe trials, all mice discriminated between the tones by reliably licking to targets while rarely licking to foils, exhibiting expert performance despite their variable and often poor performance in the presence of the licktube (Fig. 1f, 'probe context', Fig. 1g, Supplementary Movie 3, hit rate: $93.0 \pm 2.1\%$, false-alarm rate: $19.0 \pm 3.5\%$). The improvement of behavioral performance was specific to the probe context, and did not drive improvements in performance in reinforced trials immediately following the probing (Fig. 1h–j). Mice therefore appeared to understand the task contingencies many days before they expressed this knowledge in the presence of reinforcement.

We then tracked probe learning trajectories throughout learning in a subset of mice (Fig. 1k–m). Differences in acquisition versus expression were particularly acute early in learning (Fig. 1k, example mouse; Fig. 1l, summary of all mice, reinforced trials to expert: $4728 \pm 647$ trials; probe trials to expert: $1765 \pm 108$ trials; $N = 7$ mice, $p = 0.0055$). Interestingly, behavioral performance in the probe context more judiciously separated the stages of associative learning as shown by the hit and false alarm rates over learning (Fig. 1m). Animals discriminated poorly early in learning (trials 0–500) in both contexts, with a markedly lower action rate in the probe context (Fig. 1m, trials 0–500: reinforced hit rate: $82.3 \pm 3.8\%$, reinforced false-alarm rate: $78.2 \pm 2.8\%$; probe hit rate: $35.8 \pm 7.8\%$, probe false-alarm rate: $25.1 \pm 6.9\%$, $N = 7$ mice, $F(3, 18) = 33.17$, $p < 0.001$ between contexts, $p > 0.05$ within contexts, one-way repeated-measures ANOVA followed by Tukey's post-hoc correction; Fig. 1l probe context: $d' = 0.3 \pm 0.2$; reinforced context: $d' = 0.2 \pm 0.1$; $t(6) = 0.7055$, $p = 0.51$, Student's paired two-tailed t-test). Moreover, hit and false alarm rates were equally affected by the presence of reinforcement at this early stage ($\Delta$Target $= 46.6 \pm 6.8\%$, $\Delta$Foil $= 53.1 \pm 8.3\%$, $t(6) = -0.988$, $p = 0.36$, Student's paired two-tailed t-test). As learning progressed in the probe context, animals first acquired a generalized tone-reward association. This resulted in a modest increase in both the hit and false-alarm rates in the probe context (Fig. 1m). Subsequently, performance in the probe context rapidly improved as the tone-reward association became increasingly stimulus specific. Overall, these data show that the acquisition of task knowledge or contingencies (e.g., some stimuli predict positive outcomes, others do not) can be dissociated from the expression of that knowledge (e.g., the decision to lick or not).

Learning studies often focus on single task structures and single animal models, making it difficult to distill general principles of learning across species and behavior. In particular, using licking as the operant response is a potential confound, as the motor action of licking is used as both the learned motor action and the consummatory appetitive response. Moreover, head-fixed mice may use different strategies and/or be particularly sensitive to reinforcement given their limited ability to forage (due to head-fixation). For example, freely-moving rodents may engage in different types of exploratory foraging than head-fixed animals. To address whether testing context influences performance in other task structures and other species, we performed additional studies in mice, rats, and ferrets.

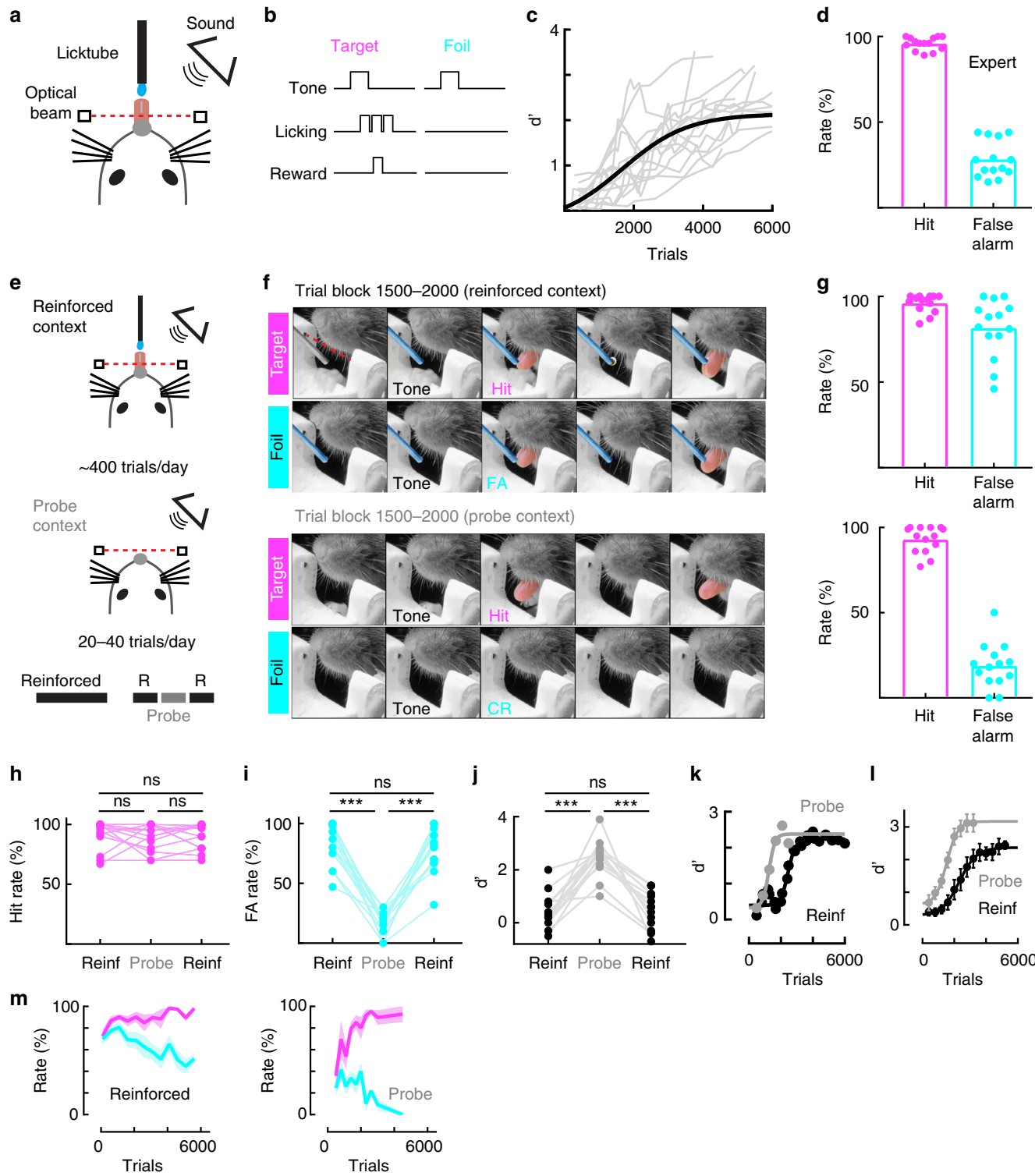

First, we tested whether separating the motor action from the consummatory response would retain (or abolish) the dissociation between task acquisition and expression. In the reinforced context, head-fixed mice were trained to press a lever in response to the target tone to gain access to water reward provided through a licktube (Supplementary Fig. 1a, b). This design added an additional key feature: the licktube was normally absent in the reinforced context and was only introduced for a short period to deliver the water and then immediately retracted. As a result, the sensory environment in the probe and reinforced contexts were

identical, removing the possibility of the licktube presence in the reinforced context as an impulsive driver of licking; instead, the possibility of reinforcement was more abstract. Early in learning (trial block 1500–2000), we found that mice pressed the lever at high rates for both the target and foil tones in the reinforced context (Supplementary Movie 4; Supplementary Fig. 1c, hit rate = 95.3 ± 3.3%, false-alarm rate = 80.0 ± 7.1%, $p = 0.3$). In the probe context, however, we observed a high response rate for the target tone but a stark reduction in responding to the foil tone, similar to what we observed in the lick-version of this task

**Fig. 1** Expression of underlying task knowledge is context-dependent. **a** Behavioral schematic in the reinforced context. **b** Mice are trained to lick to the target tone for water and to withhold from licking to the foil tone. **c** Behavioral sensitivity (d′) over trials in the reinforced context (max d′: 2.7 ± 0.2, N = 14 mice, black line is sigmoidal fit to average). **d** Hit and false-alarm rates of individual animals at peak performance rates (hit rate: 96.0 ± 0.9%, N = 14 mice; false-alarm rate: 28.0 ± 2.9%). **e** Top: same as **a**. Middle: 20–40 probe trials are interleaved with reinforced training each day, during which the licktube was removed and no reinforcements available. Bottom: typical daily training structure, 200–300 reinforced trials, 20–40 probe trials, 70–200 reinforced trials. **f** Still-frames from a movie in a training session between trials 1500–2000. Reinforced context (licktube present); mouse correctly responding to a target tone; mouse erroneously responding to a foil tone. Probe context (no licktube present, same session): mouse correctly responding to a target tone in the probe context with a lick; mouse correctly withholding a response to a foil tone. **g** Top: average hit rate (95.8 ± 1.4%, N = 14 mice) and false-alarm rate (81.6 ± 4.6%) across trials 1500–2000 in the reinforced context. Bottom: average hit rate (92.8 ± 2.1%, N = 14 mice) and false-alarm rate (18.7 ± 3.5%) in the probe context. $F_{(3,39)}$ = 198.05, one-way repeated-measures ANOVA, Tukey's post-hoc correction, p = 0.84 between hit rates, p < 0.05 for all other comparisons. **h** Hit rates remain constant across contexts (Trials 1500–2000, hit rates, pre-reinforced = 91.3 ± 3.21%, probe = 89.1 ± 2.92%, post-reinforced = 89.7 ± 2.90%, n = 14 animals; $F_{(2,26)}$ = 0.1932, p = 0.851 between pre-reinforced and probe contexts, p = 0.914 between reinforced contexts, p = 0.973 between post-reinforced and probe contexts, one-way repeated measures ANOVA, Tukey's post-hoc correction). **i** False-alarm rate during probe trials is significantly lower than during the reinforced trials (Trials 1500–2000, pre-reinforced false-alarm rate: 81.9 ± 4.5%, probe false-alarm rate: 15.5 ± 2.7%, post-reinforced false-alarm rate: 77.7 ± 4.9%, n = 14 animals; $F_{(2,26)}$ = 107.8, p < 0.0001 between reinforced sessions and probe contexts, p = 0.548 between reinforced contexts, one-way repeated measures ANOVA, Tukey's post-hoc correction). **j** Behavioral sensitivity (d′) is significantly higher during probe trials than in the reinforced sessions (Trials 1500–2000, pre-reinforced d′: 0.40 ± 0.18, probe d′: 2.46 ± 0.18, post-reinforced d′: 0.53 ± 0.19, N = 14 mice; $F_{(2,26)}$ = 44.56, p < 0.001 between reinforced sessions and probe contexts, p = 0.827 between reinforced contexts, one-way repeated measures ANOVA, Tukey's post-hoc correction). **k** Learning trajectories of an individual animal in the reinforced (black, n = 24 training sessions) and probe (gray, n = 6 training sessions) context. Dots indicate individual training sessions; lines indicate a sigmoidal fit. **l** Average d′ of a subset of animals whose learning was tracked in both the reinforced (black, N = 4–7 mice per time bin) and probe (gray, N = 4–7 mice per time bin) contexts. Dots indicate trial bins; solid lines indicate a sigmoidal fit. Reinforced trials to expert: 4728 ± 647 trials; probe trials to expert: 1765 ± 108 trials; N = 7 mice, t(6) = 4.359, p = 0.0055. **m** Average learning trajectories in the reinforced context (left) and probe context (right) (N = 7 mice, magenta = hit rate, cyan = false-alarm rate, all error bars indicate mean ± s.e.m)

(Supplementary Movie 5; Supplementary Fig. 1c, hit rate = 86.5 ± 5.4%, false-alarm rate = 37.5 ± 7.8%, p = 0.014). Later in learning, we observed high hit rates and low false-alarm rates in both the reinforced and probe contexts (Supplementary Fig. 1d). This demonstrates a clear dissociation between acquisition and expression and is similar to our observations in the previous lick-based version of this task.

Second, we assessed whether task acquisition and expression were dissociated in freely-moving rats using a different audio-visual behavioral paradigm. In the reinforced context of this Pavlovian discrimination task, a tone alone (S+) predicts the appearance of food in a food cup, whereas a light presented 5 s before the same tone (S−) reverses the tone's predictive quality (i.e., no food is delivered). In the probe context, food was not delivered to the food cup for either the S+ or S−. This task benefits from an analog measure of performance: rather than a binary decision (such as lick or no-lick), responses to each stimulus type was recorded as the percentage of the food-sampling window (5 s post-stimulus) rats spent in the food cup. We found that rats learned this task in the reinforced context within 8 trial blocks (Fig. 2b, 3–8 trial blocks to expert). Remarkably, freely-moving rats reliably discriminated in probe trials much earlier in training than they did in reinforced trials—all rats spent significantly less time at the food cup following the S− stimulus than following the S+ stimulus (Fig. 2c). Thus, paralleling the task-learning in head-fixed mice, probe trials revealed that rats in this freely-moving task had acquired the correct stimulus-action associations long before their performance in the presence of reinforcement reached expert levels (Fig. 2d). Animals also performed significantly better in the probe context even after their performance had reached expert levels in the reinforced context (Fig. 2e).

Third, we aimed to determine whether dissociation between acquisition and expression could be observed in a fear conditioning task, and therefore examined the behavior of rats in a previous study (Holland and Lamarre, 1984[22]). In this discrimination task, rats were first trained to press a lever for sucrose reinforcement, and Pavlovian fear conditioning procedures were subsequently superimposed on this operant lever pressing baseline. When a tone target stimulus was presented alone (S+), it was paired with foot shock, but not when it was presented following a light feature (S−) (Supplementary Fig. 2a). Fear conditioning was assessed by measuring the suppression of operant lever press responding during the tone, and discrimination as the difference in suppression ratio to the S+ and S−. In this earlier study, Holland and Lamarre[22] assessed performance in reinforced versus probe contexts but did not explicitly compare the two. We found that—similar to our results with appetitive conditioning—performance in the probe context was greatly improved compared with the reinforced context in trained animals (Fig. 2f). In the probe context, animals did not change their response to the S+, but significantly increased lever presses following S− (Supplementary Fig. 2b). Thus, in comparison with reward learning tasks, animals increased behavioral responses in the probe context to more accurately discriminate between the target and foil stimulus.

Fourth, we tested whether the dissociation between reinforced and probe contexts could be observed in freely-moving animals when the motor action was distinct from the consummatory response. In this operant ambiguous-feature discrimination task[23], a single light feature stimulus indicated both that sucrose reinforcement was available for lever pressing during a target tone, and that reinforcement was not available during a white noise stimulus. Thus, the rats were trained with a discrimination procedure in which lever presses were reinforced during a light + tone compound, but not during that tone alone, and during a noise when it was presented alone but not during a compound of light and noise. Because each stimulus (light, tone, and noise) is ambiguously related to reinforcement, a more complicated form of configural learning is required to solve the task. In this task, we also observed a significant improvement in performance in the probe compared with the reinforced contexts (Fig. 2g, h).

Finally, we ensured that these results were not specific to rodents, by performing similar behavioral experiments in two ferrets. Ferrets are carnivores with gyrencephalic brains and well-differentiated frontal cortices similar to primates[24]. We trained two head-fixed ferrets to discriminate between two click-trains in a go/no-go task design. Ferrets also performed substantially better in the probe

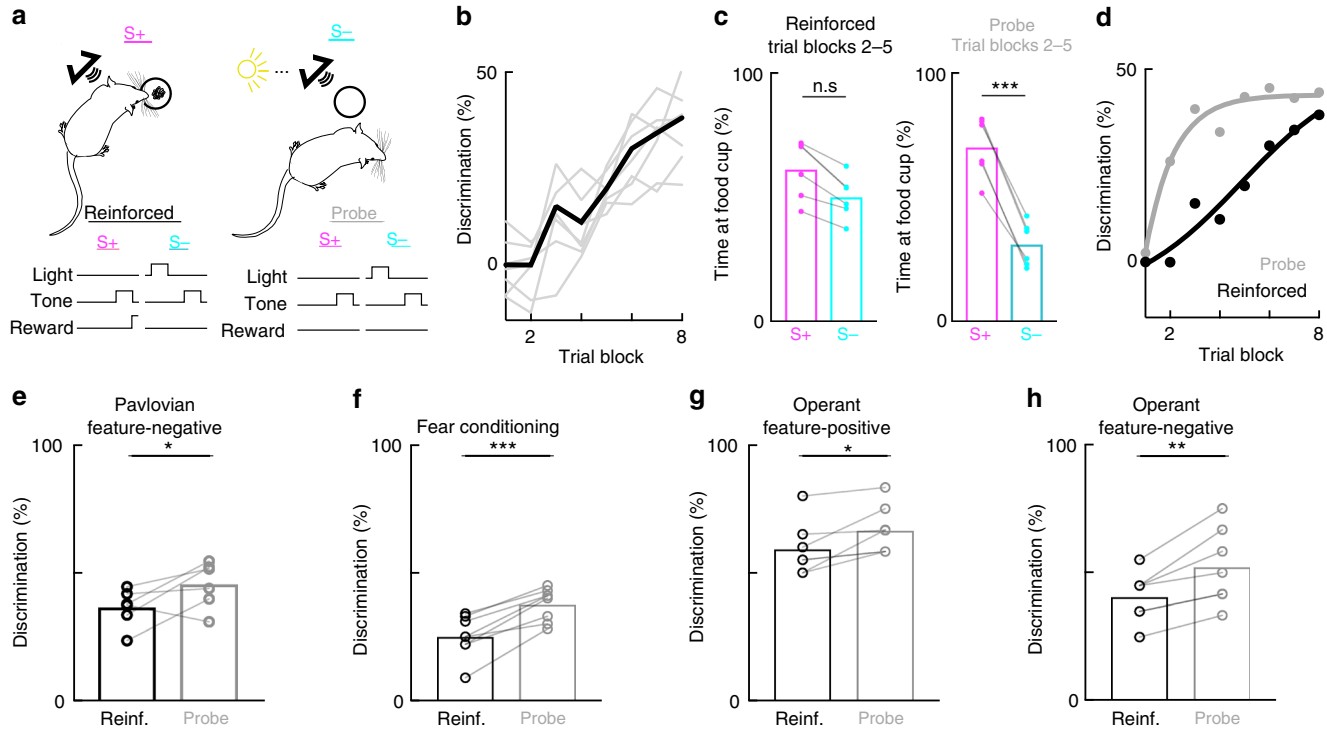

**Fig. 2** Dissociation of acquisition and expression generalizes to freely-moving rats. **a** Behavioral schematic of the discrimination task. **b** Discrimination between stimuli as a function of trial blocks in the reinforced context; gray lines indicate individual animals, black line is the average performance across animals (peak discrimination: 35.5 ± 3.8%, N = 6 rats). **c** Left: average time at food cup after S+ stimulus (61.2 ± 4.8%, N = 6 rats) and S− stimulus (50.1 ± 3.5%) across trial blocks 2–5 in the reinforced context. Right: average time at food cup after S+ stimulus (70.1 ± 4.9%, N = 6 rats) and S− stimulus (31.0 ± 3.6%) across trial blocks 2–5 in the probe context. F(3,20) = 15.49, one-way repeated-measures ANOVA followed by Tukey's post-hoc correction, $p = 0.84$ between S + rates, $p = 0.39$ between reinforced S+ and S− rates, $p < 0.05$ for all other comparisons. **d** Average discrimination of animals in the reinforced and probe contexts over learning. Dots indicate experimental data averaged across rats, lines are the least-squares sigmoidal fit.
**e** Discrimination of rats fully trained (trial block 8) on the Pavlovian feature-negative discrimination task in **a**. Rats discriminated significantly more between the S+ and S− during probe trials (Reinforced discrimination: 33.8 ± 3.4%, N = 6 rats; Probe discrimination: 43.2 ± 3.7%; two-tailed Student's paired t-test, t(6) = 2.693, $p = 0.036$). **f** Discrimination of rats in a fear-conditioning based discrimination task. All rats discriminated significantly more between stimuli in the probe context than in the reinforced context (Reinforced discrimination: 25.1 ± 2.4%, N = 8 rats; Probe discrimination: 37.6 ± 3.1%; two-tailed Student's paired t-test, t(7) = 7.349, $p = 1.56 \times 10^{-4}$). **g** Performance of rats in an operant ambiguous feature discrimination in reinforced and probe contexts of the feature positive portion of task. Rats discriminated significantly better in the probe context than in the reinforced context (Reinforced discrimination: 59.2 ± 4.0%, N = 7 rats; Probe discrimination: 66.6 ± 3.6%; two-tailed Student's paired t-test, t(6) = 3.15, $p = 0.020$). **h** Performance of rats on the same task as **g** on the feature negative discrimination portion (i.e., light + noise versus noise alone). Rats discriminated significantly better in the probe context than in the reinforced context (Reinforced discrimination: 40.71 ± 3.7%, N = 7 rats; Probe discrimination: 52.4 ± 5. 7%; two-tailed Student's paired t-test, t(6) = 4.534, $p = 0.004$). All error bars indicate mean ± s.e.m

context much earlier in training as compared with the reinforced context (Supplementary Fig. 3a–c). Taken together, the dissociation between acquisition and expression reveals latent knowledge in mice, rats, and ferrets and across a variety of task designs.

**A network model captures the learning dynamics**. What computational mechanisms may underlie the dissociation between learning curves in reinforced and probe trials? Classical reinforcement learning theory describes behavioral learning in terms of two systems, one that updates values of different stimulus-action associations based on the obtained reinforcement, and another that generates actions in response to stimuli by reading out the values of the different options. We hypothesized that learning of action values takes place only during reinforced trials, while the changes between contexts (reinforced and probe trials) do not change the learned values of different options, but modulate only the read-out parameters to consider factors such as impulsivity or exploration. Such a mechanism would lead to a difference at the level of behavioral performance between contexts, without any change of the underlying action values which represent task knowledge.

To test this hypothesis, we focused on a specific network implementation of reinforcement learning for go/no-go tasks[17,25]. We constructed a computational model of reinforcement learning in which action values were represented at the level of synapses projecting from a sensory population (S+, S− and S) to output populations (D and I) (Fig. 3a, gray), while action generation was governed by the parameters of the upstream readout units (Fig. 3a, orange). The sensory population consists of two tone-selective populations representing target (S+) and foil (S−) stimuli, and one population that is tone-responsive but has no preference for stimulus-identity, consistent with the functional organization of auditory cortical networks[21,26–29]. The sensory populations project to an inhibitory population (I) and an excitatory read-out population (D) with plastic synapses in the decision-making area. This type of model is biologically plausible and has been found to more accurately characterize rodent behavioral data than standard reinforcement learning models[17]. In our model, the equation by which the readout units processed information from the sensory population was changed in a context dependent fashion by way of a single parameter. We fit the model to our mouse, rat and ferret data and examined

                    5

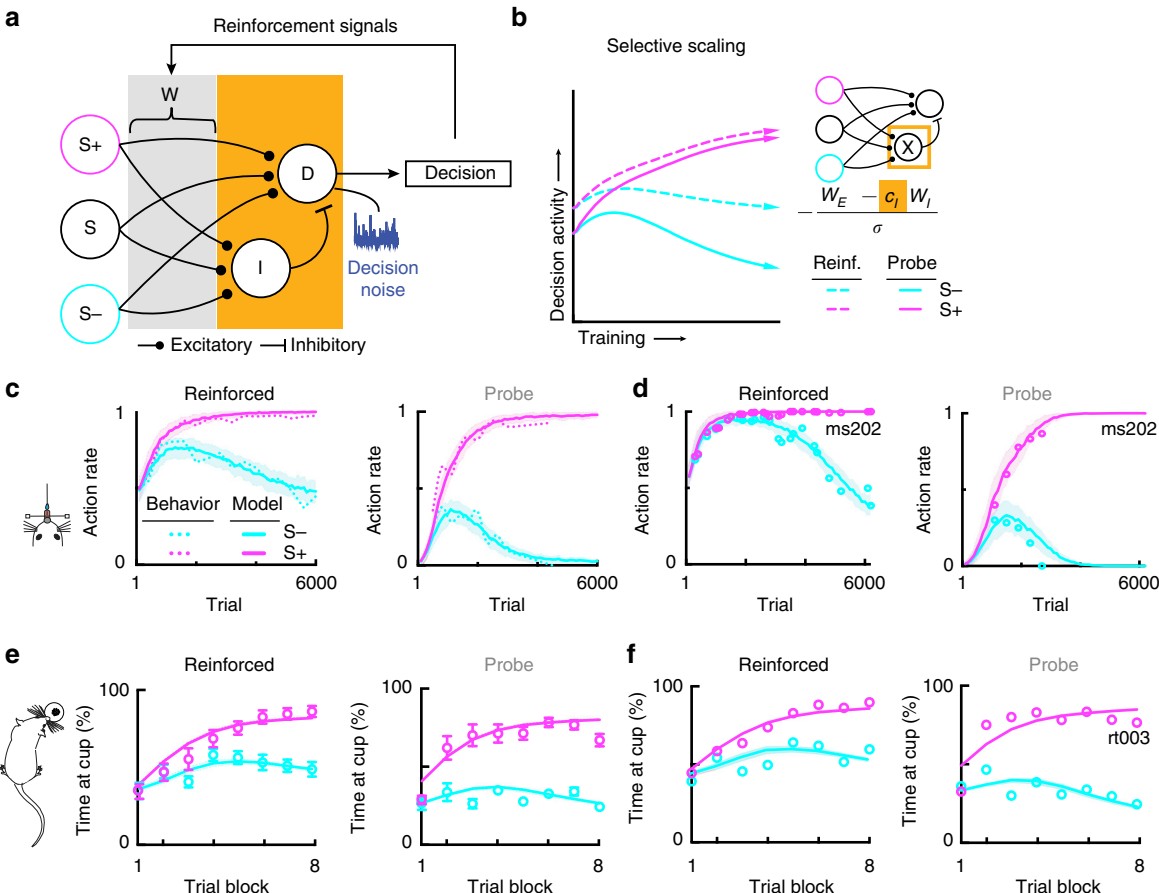

**Fig. 3** Contextual scaling in a reinforcement learning model. **a** Schematic of the model, which implements reinforcement-driven learning of stimulus-action associations, with a readout function that can be contextually modulated. The model simultaneously captures reinforced and probe learning trajectories by dissociating between reward-driven plasticity representing task acquisition (gray, reinforcement signals) and context-dependent changes in expression (orange, contextual scaling) of the learned values. Plastic synapses between sensory and decision-making populations represent stimulus-action values, and their weights are only updated during reinforced trials (gray shading, reinforcement signals). Actions are generated by the decision-making (D and I) units (orange shading), which read out the sensory input filtered through the synaptic weight matrix (W). The decision unit's (D) activity determines the probability that the model will respond. The parameters of the readout units (orange shading) are modulated between the reinforced and the probe contexts, either via selective scaling of inhibition and excitation, noise modulation, or threshold changes (see Supplementary Fig. 4). **b** Left: illustration of the effects of selective scaling of inhibition on D's activity. Decision activity represents the net input to the decision-making unit, i.e., the difference between the values of the go and no-go actions, for the target (magenta) and foil (cyan) tone over the course of learning. Solid and dashed lines, respectively, indicate probe and reinforced context (inhibition scaling $c_I$: 0.52). Right: schematic of the contextually modulated model via inhibitory scaling, and a description of how inhibitory scaling is mathematically implemented. The net output of the model is proportional to the excitatory weights ($W_E$) minus the inhibitory weights ($W_I$). In the probe context, the inhibitory weights are multiplied by a scalar factor ($C_I$, orange). Throughout panels a-b, orange highlights the context-dependent parameters. **c** Comparisons between average mouse behavioral data ($n = 7$ mice) and fits of the inhibitory scaling model for the two contexts. **d** same as **c** but for the learning trajectory of one individual mouse. **e**, **f**, same as **c**, **d**, but for rat behavioral data (**e**: $n = 6$ rats; **f**: rat rt003)

whether contextual modulation of the readout could quantitatively account for the behavioral learning trajectories (Fig. 3c–f).

We found that our model simultaneously captured the learning curves in the reinforced and probe trials each of the tasks across mice, rats, and one ferret. This minimal model therefore provided a parsimonious description of a large and diverse dataset (Fig. 3c–f, Supplementary Fig. 3d), and validated our hypothesis that changes between contexts only modulate how the learned values of stimulus-action associations are read out, but not the values themselves (Fig. 3b). The model moreover constrained the possible mechanisms underlying the contextual modulation of the readout. One possibility was that context modulated only the behavioral readout via scaling of the readout gain that classically determines the amount of exploration[30,31]. This candidate mechanism accounted poorly for the behavioral data (Supplementary Fig. 5a), largely because of its symmetry between target and foil stimuli. This symmetry ensured that if the false-alarm rate was greater than 50%

in the reinforced context in a given session, the false-alarm rate could not be below 50% in the probe context, inconsistent with the behavioral data (Supplementary Figs. 4a and 5a, *reinforced* and *probe*). A second possibility was an additive modulation equivalent to a threshold shift (Supplementary Fig. 4b)[31]. While this mechanism provided a better fit to the data, it still did not simultaneously capture the trajectories in both contexts (Supplementary Fig. 5b, *reinforced* and *probe*), as again the readout function for target and foil trials was affected in a highly correlated manner (Supplementary Figs. 4b and 5b). A third approach was to scale independently the drive for no-go or go responses by modulating either the gain of inhibition (Fig. 3b, Supplementary Figs. 4c and 5d), or the excitatory drive to the decision unit (Supplementary Figs. 4d and 5c). Interestingly, selectively scaling the gain of feed-forward inhibition provided the best fit of behavioral data with a small number of adjusted parameters for mice, rats, and ferrets (Fig. 3c, d, Supplementary Figs. 3e, 9f and

10f). This straightforward mechanism for selectively scaling the no-go response is absent in classical reinforcement learning models yet quantitatively describes the dissociation between acquisition and expression of task knowledge.

One alternative computational explanation in the case of the head-fixed mice is that the licktube-reward association supersedes the target-reward association; this would predict continuous or random licking. However, early in learning, baseline lick rates in the reinforced context were low, with a robust increase in licking after the tone (Supplementary Fig. 6a, b). Theoretically, these effects could also be mediated by a compound association, whereby the licktube provides an additive drive to lick, bringing animals closer to an internal response threshold, even if baseline lick rates are low. If this were the case, for reinforced sessions in which the hit and false-alarm rates were both below 100%, we would expect that removal of the licktube would equally reduce hits and false-alarms (i.e., subtracting the additive drive to lick by the licktube). This was not the case; our behavioral data show that false alarm rates were significantly more affected than hit rates by context switching (Supplementary Fig. 6c, d; $\Delta$Target $= 14.9 \pm 18.7\%$, $\Delta$Foil $= 48.6 \pm 18.7\%$, $p = 2.79 \times 10^{-5}$). Moreover, in the experiments with freely-moving rats, the food cup was always present with only the reinforcer (i.e., food pellets) either being present or absent, and in lever-pressing tasks for both mice and rats the motor action was separated from the consummatory action. This largely negates the possibility of a compound association as the likely mechanism.

**Testing context strongly drives inter-individual variability**. One challenge in evaluating behavioral data and building robust learning models is that learning curves appear highly variable across individual animals[16,17] and humans[32]. Typically, this variability has been thought to arise from differences in how quickly animals learn stimulus-action associations; "smarter" animals make associations faster, represented in formal reinforcement learning models via parameters related to reward-based plasticity. We examined individual animal learning trajectories in animals in which we collected both reinforced and probe behavioral performance consistently during learning ($N = 7$ mice, $N = 6$ rats). We found that selective scaling of the decision readout could capture parallel behavioral trajectories of individual mice and rats with high fidelity (Fig. 3d–f, Supplementary Figs 9 and 10). As expected, mice exhibited significant behavioral variability in how quickly they reached expert levels in the reinforced context (Fig. 4a, left panel). Surprisingly, in the probe context, this variability was strongly suppressed, revealing that different animals had acquired task knowledge at nearly identical rapid rates (Fig. 4a, right panel). We quantified this by calculating the number of trials it took mice to reach expert performance and the variance of this between animals (Fig. 4b, $d' > 2.0$ with false-alarm rates <50% for 100+ trials). Probe learning trajectories were stereotyped across animals while reinforced learning trajectories were much more variable (Fig. 4b). For rats, the inter-animal variability in learning rates was also much lower in the probe context than in the reinforced context (Fig. 4c, d) further emphasizing the generalizability of our findings across species.

We tested in our model whether the inter-individual variation in performance was primarily explained by variability in reward-based plasticity parameters or variability in contextual scaling of the decision readout. To do so, we utilized a one-factor-at-a-time approach to examine how much each parameter could alter the learning curve versus how much real learning curves differed, with each parameter constrained by individual animal fits. First, we established the average parameters required to fit the average behavioral data (9 parameters). Next, we varied a single

parameter (i.e., $c_I$) within the range corresponding to all of the individual animal fits, and calculated the resulting error relative to the average fit for each value of the parameter (RMSE with respect to average fit). We found the maximum error generated by this entire range of values (Maximum Model Error). We then calculated the maximum error within the behavioral data (Maximum Behavioral Error; RMSE of individual learning trajectories with respect to average learning trajectories), and defined explained variation as $\frac{\text{Maximum model error}}{\text{Maximum behavior error}}$. Finally, we performed this calculation for each of the model parameters. To determine how different parameters contributed to the model error, we divided our parameters into four groups: learning rates ($\alpha$, $\alpha_{NR}$), initial conditions ($W_E$, $W_I$, $W_{SE}$, $W_{SI}$), noise ($\sigma$), and inhibitory scaling ($c_I$). To remain conservative in our analysis, the parameter in each group that explained most variation was selected to be representative. Interestingly, the contextual scaling of inhibition could explain nearly all of the variation in performance in the reinforced context while reward-based plasticity parameters (learning rates, initial conditions, and noise) were less explanatory (Fig. 4e, f). Individual performance variance therefore appears to emerge more from contextual factors than from differences in underlying rates of associative learning.

## Discussion

In the 1930s, Edward Tolman and colleagues elegantly demonstrated that the introduction of reinforcement can critically mediate the generation and expression of a "cognitive map"[33,34]. Since its inception, Tolman's cognitive map hypothesis has profoundly impacted how neuroscience and behavioral psychology think about and approach cognitive behaviors. Here, we show that this simple yet powerful behavioral manipulation (introduction and removal of reinforcement), which can dissociate between the acquisition and expression of sensorimotor task knowledge during learning. Across a wide range of behavioral tasks and animal species, we demonstrate that the apparent lack of discrimination between two conditioned stimuli early in learning can be attributed to contextual factors rather than underlying knowledge. Access to reinforcement masked the ability to execute correct stimulus-action associations, which can be revealed simply by testing animals in a different context where the reinforcement is absent. This hidden learning appears to be faster and highly stereotyped across animals, indicating that apparently-robust inter-individual differences in the presence of reinforcement are not driven by inter-individual differences in sensorimotor abilities for these task designs.

In these sensorimotor behaviors, the acquisition of task knowledge likely operates via reward-based plasticity from a sensory to decision-making population. These projections rapidly stabilize and enable discrimination between the action values of the stimuli. Interestingly, neural data acquired during learning suggests that perhaps this rapid learning of stimulus-action associations may be reflected in sensory cortex. In the primary visual cortex of mice, for example, neural sensitivity to trained stimuli increases well before behavioral improvements[13]. These behavioral measurements, however, were performed in the testing context suggesting that an alternate measure of behavior, such as our probe context, may have shown that the neuronal sensitivity tracks sensorimotor task acquisition (i.e., probe context learning rate) but precedes task expression (i.e., reinforced context learning rate) in the testing context. Thus, rapid changes in V1 may reflect core task learning while performance-correlated neural changes observed in other studies[10,35] reflect a more complex mix of contextual factors including behavioral state and cost-benefit considerations. The relative timing of neural changes versus behavioral improvements has profound implications for neural

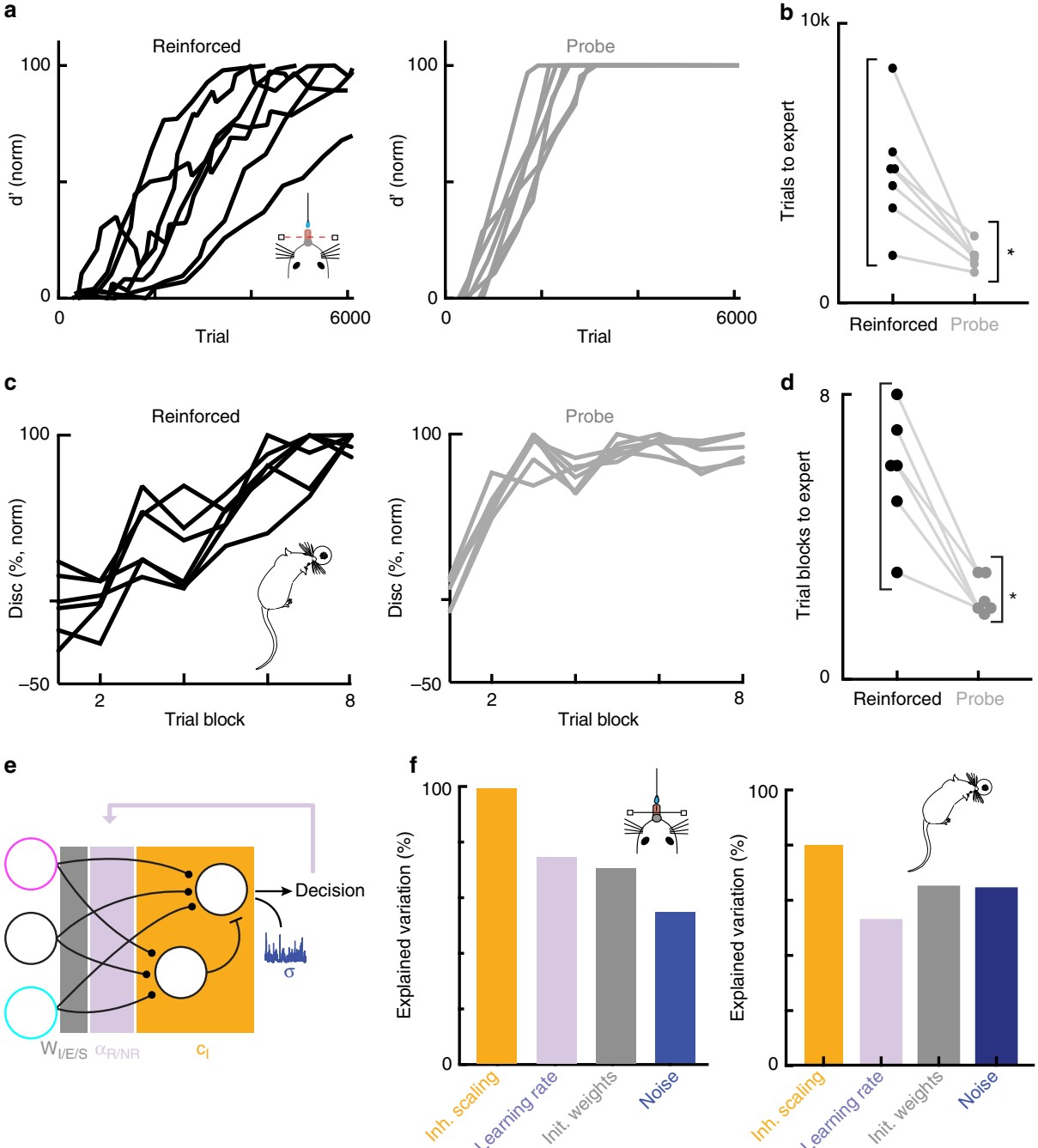

**Fig. 4** Performance variance across rodents arises from contextual factors. **a** Left: normalized discrimination of all mice in the auditory go/no-go task over the over the course of learning ($N = 7$ mice) in the reinforced context; right: same as left but in the probe context ($N = 7$ mice). **b** Number of trials required for individual animals to reach expert performance levels ($d' > 2.0$, false-alarm rate <50% for at least 100 trials) in the reinforced (black) and probe (gray) contexts. The number of trials required for expert performance is significantly more stereotyped across animals in the probe context than in the reinforced context (probe: st.d. = 358 trials, range = 1348–2332, $n = 7$ animals; reinforced: st.d. = 2047 trials, range = 2187–8838 trials, $N = 7$ mice, $F_{(6,6)} = 32.63$, $p = 5.025 \times 10^{-4}$, two-tailed two-sample F-test for equal variances). **c** Left: normalized discrimination of rats in the Pavlovian feature-negative discrimination task over the over the course of learning ($N = 6$ rats) in the reinforced context; right: same as left but in the probe context (N = 6 rats). **d** Number of trials required for individual animals to reach expert performance levels (discrimination >25% for at least 1 day of training, i.e., 16 trials) in the reinforced (black) and probe (gray) contexts. The number of trials required for expert performance is significantly more stereotyped across animals in the probe context than in the reinforced context (probe: st.d. = 0.51 days, range = 2–3 days, $N = 6$ rats; reinforced: st.d. = 1.72 days, range = 3–8 days, $F_{(5,5)} = 11.12$, $p = 0.0194$, two-tailed two-sample F-test for equal variances). **e** Schematic of network model with colors indicating parameter groups. **f** Left: Percentage of inter-individual variation for mice performing the auditory go/no-go task explained by the four core model parameters: inhibitory scaling: 99.7%; learning rates: 74.8%; initial weights: 70.8%; noise: 54.6%. Right: same as left but for modeling of rats learning the Pavlovian feature-negative discrimination task (variation explained by inhibitory scaling: 80.7%; learning rates: 53.0%; initial weights: 65.2%; noise: 60.6%)

models of learning and the underlying neural implementation, particularly as it relates to how specific brain structures instruct versus permit plasticity[36,37]. Similar re-interpretations of existing results[6,11,12] may help us improve our understanding of the population-level and single-neuron dynamics during learning across sensorimotor regions.

The circuit mechanism of reward-based plasticity of sensorimotor projections is a critical area for future exploration. Dopaminergic neurons have been implicated in calculating reward prediction error[38] yet whether and how these signals propagate to sensory cortices remains unclear. One possibility involves the entrainment of a neuromodulatory system with broader cortical projection patterns, such as cholinergic projection neurons in the basal forebrain[39]. Recent work in trained animals suggests that the cholinergic projection systems signals reinforcement feedback in a phasic manner[40]. Future studies will explore how phasic cholinergic signals are involved in learning-related plasticity and the forms of behavioral changes documented here.

Our model further suggests that task expression is influenced by contextual scaling of the decision-making population. What contextual factors might be responsible for this? Some context-dependent responses may be maladaptive; for example, impulsivity or anxiety may hasten the response function by reducing inhibition under motivated conditions. Interestingly, animals responded more quickly in the reinforced context (Supplementary Fig. 7) suggesting that impulsivity may be one such contextual factor. Others may be adaptive such as increased foraging and exploration early in learning in the presence of reinforcement, which in the case of our mouse task would drive an increase in false alarm rate. Computing the reward rate shows that non-discriminant licking in the reinforced context to both target and foil tones maximizes reward at early stages of training (Supplementary Fig. 8). Thus, both adaptive and maladaptive factors may contribute to context-dependent expression.

What might be the neural implementation of this contextual scaling? The state-dependent nature of the behavioral transitions and the potential role of inhibition suggest that neuromodulation, e.g., acetylcholine or noradrenaline, may be involved. Moreover, prefrontal mechanisms of top-down control may also play a role in stabilizing behavior in the presence of reinforcement. Behaviorally isolating the underlying learning rates and drivers of variability, however, will be critical if we want to link behavioral output, the computational algorithms that enable this output, and the relevant neural implementations[4,41].

More broadly, the dissociation between knowledge and expression has critical implications for how we understand the distributed computations that enable learning. The possibility that acquisition and expression rely on different physiological mechanisms may help us isolate the root causes of under-performance and variability in learning rates. This dissociation may be particularly relevant when reward, motivation, stress, and arousal are high and intermingled. Regardless, behavioral and theoretical dissociation of acquisition and expression of knowledge in learning now provides us with a conceptual framework to better explore the neural basis of individual variability. The possibility of distinct mechanisms between acquisition and expression may help us identify the neural basis for performance variability across a wide range of behavioral, perceptual, cognitive, and intelligence testing contexts, including possibly in humans.

## Methods

**Animals.** All mice procedures were approved under a New York University IACUC protocol and a Johns Hopkins University IACUC protocol. Male and female mice of mixed sex were used at 8–16 weeks of age. Multiple strains were used (C57/BL6, PV-cre, ChAT-ChR2). Behavior was a quantitative assessment with no treatment groups, and animals were thus not randomly assigned into experimental groups. The care and experimental treatment of rats was conducted according to the National Institutes of Health's *Guide for the Care and Use of Laboratory Animals*, and the protocol for "Experiment 1" was approved by an IACUC at Duke University. "Experiment 2" was conducted at the University of Pittsburgh before the establishment of IACUCs. All rats were male Long-Evans rats tested around 90 days of age. All experimental procedures involving ferrets conformed to standards specified and approved by the French Ministry of Research and the ethics committee for animal experimentation n°5. Blinding of experimenters was not relevant for this study as behavior was assessed quantitatively based on objective, measured criteria.

**Behavioral training: head-fixed mice.** All behavioral events (stimulus delivery, reward delivery, inter-trial-intervals) were monitored and controlled by a custom-written MATLAB (MathWorks) program interfacing with an RZ6 auditory processor (Tucker-Davis Technologies), and an infrared beam for lick detection. Training was initiated after surgery for head-fixation and at least 7 days of water restriction in adult mice (8–16 weeks of age, mixed sex, mixed background strain). Training was conducted during the day and began with habituation to head-fixation, which was followed by 1–2 water-sampling sessions while animals were immobilized in a Plexiglas tube facing a licktube. The licktube was typically placed at the maximal distance away from the mouse to discourage compulsive licking. Animals were then immediately placed in the complete behavioral paradigm with minimal shaping. Task training began with a 200–400 trials in the reinforced behavioral context, where we used a go/no-go auditory discrimination task with the target and foil stimuli set at 9.5 kHz or 5.6 kHz (stimuli-salience pairing randomly assigned, 0.75 octave spacing). Target versus foil trials were pseudo-randomly ordered, each of which consisted of a pre-stimulus period (1.25 s), stimulus period (100 ms), delay (50 ms), response period (1.75 s), and an inter-trial interval (ITI) with variable duration as described below.

Tones were presented to animals under two different behavioral contexts. In the 'reinforced context', a licktube delivering water was positioned within tongue reach (0.5–1.0 cm). In this context, mice only received water for correct licks to the target tone during the response period. Incorrect licks during the response window to the foil tone (a false-alarm) resulted in a mild negative punishment consisting of an extended ITI. Animals were not punished if they licked during any other time epoch (i.e., if animals licked in the pre-stimulus period, tone presentation or delay period, the trial continued with the standard ITI). This enabled us to confirm that animals were actively increasing lick rate for target tones during hit trials and reducing lick rate for foil tones during correct reject trials. This measurement confirmed that both the target and foil tone had behavioral effects on the animal; without this, animals could take a single-tone strategy (i.e., learn to lick only for the target tone or withhold licking for the foil tone). Hit trial ITIs were 4–5 s (to enable licking for full reward), miss trials were not punished and had an ITI of 2–3 s, false-alarm trials were punished with an ITI lasting 7–9 s, and correct rejects immediately moved to the next trial with an ITI of 2–3 s. In the second context, the 'probe context', the licktube was removed from the behavioral space by an automated actuator, such that it was out of sight and whisker reach. Target and foil trials were again presented in a pseudo-random order, but did not correlate to the presence of potential rewards or punishments. We continued to monitor behavioral responses made during the response period following stimulus presentation, but trial durations were not dependent on such behavioral responses (ITI ~2–3 s).

Each day, animals were typically trained on two blocks of trials in the reinforced context (100–300 each, total of ~400 reinforced trials per day), and one randomly interleaved block of probe trials (20–40 trials). Importantly, because mice were not presented with any direct incentives to execute behavioral responses in the probe contexts, the number of trials in probe blocks could not be further extended, as this caused rapid cessation of behavioral responses. Utilizing the short probe blocks, behavioral responses (hit rate and false-alarm rates) in the probe context (i.e., in the absence of reinforcement) begun to decline after 60–150 total passive trials across 3–6 days of training, as animals learned that the absence of the licktube indicated an absence of reinforcement. For a subset of mice ($N = 4$), we introduced probe blocks more sparsely (~1 probe block per 1500 reinforced trials) to ensure that the decline of behavioral responses in the probe context occurred independently from the training in the reinforced context.

Average performance in the reinforced condition was measured by segmenting performance into discrete blocks, such that we averaged all false-alarm and hit rates recorded in blocks of 100 trials. A similar process was utilized for measurements of average performance in the probe context, but because of the smaller number of probe blocks, block sizes were adjusted to ensure that probe blocks from at least two animals were incorporated into each measurement (max 500 trials/block). For individual animals, reinforced performance was again averaged in blocks of 100 trials, and probe performance was averaged across individual probe sessions. For analysis of probe learning trajectories, we only included probe blocks up until the hit rate reached a peak value, as behavioral responses in the probe context were subsequently diminished because of the absence of a positive reinforcer. Behavioral sensitivity ($d'$) to the task-relevant tones was calculated as the $z$-scored hit rate minus the $z$-scored false-alarm rate. To avoid infinite values during sensitivity calculations rates of zero and one were corrected by $\frac{1}{2N}$ and $1 - \frac{1}{2N}$, respectively, where $N$ is the number of trials in each measurement.

**Behavioral training: rats**. The subjects ($N = 7$) of the Pavlovian serial feature negative discrimination task (Fig. 2a) were male Long-Evans rats (Charles River Laboratories, Raleigh, NC, USA). Rats were housed individually in a colony room with a 4:10 h light-dark cycle. After one week of acclimation to the vivarium with ad libitum access to food and water, rats were food-restricted such that their weights reached and were maintained at 85% of their free-fed weights. Beginning three days before the first day of food restriction, all rats were handled, weighed, and fed daily until the end of the experiment. For all rats, daily behavioral testing sessions began 7 days after the beginning of food restriction, and were conducted during the light portion of the light-dark cycle. The rats were tested at about 90 days of age. The animal protocols were approved by an IACUC at Johns Hopkins University.

For the Pavlovian serial feature negative discrimination task, the behavioral training apparatus consisted of eight individual chambers ($22.9 \times 20.3 \times 20.3$ cm) with stainless steel front and back walls, clear acrylic sides, and a floor made of 0.48-cm stainless steel rods spaced 1.9 cm apart. A food cup was recessed in a $5.0 \times 5.0$ cm opening in the front wall, and photocells at the front of the food cup recorded time spent in the cup. Grain pellets were delivered to the food cups by pellet feeders (Coulbourn H14-22, Allentown, PA, USA). A jeweled 6-W signal lamp was mounted 10 cm above the food cup; illumination of this panel light served as the visual stimulus. Each chamber was enclosed inside a sound attenuating shell. An audio speaker and 6-W house light were mounted on the inside of each shell.

All rats were first trained to eat grain pellets (45 mg, Formula 5TUM, Test Diets, Richmond, IN, USA) from the food cups, in a single 64-min session, which included 16 un-signaled deliveries of 2 pellets each. In this and all subsequent session, events were delivered at random inter-trial intervals (ITIs, mean: 4 min, range: 2 to 6 min). Then, the rats received a single 64-min pre-training session, which included 16 reinforced trials; each trial consisted of a 5-s presentation of a 78 dB SPL, 1500 Hz square wave tone followed immediately by the delivery of 2 grain pellets. Finally, the rats received eight 64-min daily training sessions, each including 4 reinforced trials (S+; as before) and 12 foil trials (S−), in which a 5-s illumination of the panel light was followed, after a 5-s empty interval, by the 5-s tone, but no food delivery. The trials occurred in random order, changed daily. Four hours before each of training sessions 2–8, and 20 h after session 8, rats received a probe test, which comprised two light-tone and 2 tone-alone trials. No food was delivered in these tests. Responses to each stimulus type was recorded as the percentage of the food-sampling window (5 s post-stimulus) rats spent in the food cup. Performance, or discrimination, was recorded as the raw difference rats spent in the food cup following the S+ versus S− stimulus. Expert performance levels was defined as a discrimination >25% for at least 1 day of training, 16 trials.

The operant ambiguous feature discrimination task ("Experiment 1"; Fig. 2a–e) and the fear conditioning (conditioned suppression) in feature-negative discrimination task ("Experiment 2"; Fig. 2b) have been previously described in detail[22,23]. The subjects of Experiment 1 ($N = 7$) were male Long-Evans rats (Charles River Laboratories, Raleigh, NC, USA) and the subjects of Experiment 2 were 4 males and 4 female Sprague–Dawley rats (bred at the University of Pittsburgh). The rats in Experiment 1 received sham lesions of the hippocampus (Gallagher and Holland[23]) prior to training procedures. Rats were individually housed in a colony room with a 12:12 h light-dark cycle. All rats were carefully food-restricted to maintain 85% of their free-feeding weights, as described above. The care and experimental treatment of rats was conducted according to the National Institutes of Health's *Guide for the Care and Use of Laboratory Animals*, and the protocol for Experiment 1 was approved by an IACUC at Duke University. Experiment 2 was conducted at the University of Pittsburgh before the establishment of IACUCs. The training apparatus for these experiments was identical to the chamber-divided box described above, with sucrose solution being delivered to the food cup via solenoid valves. A $2.5 \times 2.5$ cm response lever was mounted 2 cm left of the food cup.

Experiment 1 was designed to assess performance in a discrete-trial operant ambiguous feature discrimination. A single light feature stimulus indicated both that sucrose reinforcement was available for lever pressing during a tone target stimulus and that reinforcement was not available during a white noise stimulus. Thus, the rats were trained with a discrimination procedure in which lever presses were reinforced during a light + tone compound, but not during that tone alone, and during a noise when it was presented alone but not during a compound of light and noise. Rats were first trained to consume sucrose reinforcement from the food cups and to press the lever. In the initial session, they first received 20 response-independent 0.3-mL deliveries of 6.4% (v/v) sucrose (the reinforcer used throughout this experiment) on a variable-time 1-min schedule. Each lever press was reinforced during that 20-min period and during the remaining 40 min of the session. In the next session, lever presses were reinforced, but there were no response-independent sucrose presentations; each rat was allowed to remain in its chamber until it had made about 50 lever presses. All subsequent training sessions were 60 min in duration.

The next 5 sessions were designed to establish lever pressing during the two reinforced stimuli, light (PT+) and a white noise stimulus (N+). During each of these sessions, there were 30 15 s presentations of a 73 dB SPL white noise (N) and 30 15 s presentations of a compound that comprised a 74 dB SPL 1500 Hz tone and the illumination of the panel light (PT). In the first 2 sessions, each lever press made during one of these cues was followed by sucrose delivery. In the remaining

sessions (of both this and subsequent phases), reinforcement was available only during the final 5 s of each reinforced cue. During all sessions throughout this experiment, trial sequences were generated randomly for each session. Inter-trial intervals were randomized daily, with the constraint that the range of intervals was from 0.5 to 2.0 times the mean interval (60 s).

Next, discrimination training began, in which illumination of the panel light (P) indicated the availability of reinforcement during the tone (T) and the nonavailability of reinforcement during the noise (N). All rats received four kinds of trials in each of the 20 discrimination sessions. Reinforced PT+ and N+ trials were identical to those received previously. In addition, there were 15-s presentations of the tone alone (T−), and of a compound of the panel light and the noise (PN−). In each of sessions 1–10 there were 15 of each trial type, randomly intermixed, and in each of sessions 11–20 there were 10 N+, 10 PT+, 20 PN−, and 20 T− trials. After the 20 discrimination sessions, a single non-reinforced probe test was given, which included 12 presentations of each of these trial types, plus 12 15-s presentations of *P* alone, to assess conditioning established to that stimulus.

Experiment 2 was designed to assess learning of a serial feature negative discrimination in a conditioned suppression experiment. Rats were first trained to press a lever for sucrose reinforcement feature. Pavlovian fear conditioning procedures were then superimposed on this operant lever pressing baseline. When an auditory stimulus (pure tone) was presented alone, it was paired with foot shock; when it was presented following a visual stimulus (light flash), no shock was delivered. Fear conditioning was assessed by measuring the suppression of operant lever pressing during the tone. Rats were first trained to consume the sucrose reward (0.3 ml of 8% v/v sucrose solution) from the food cup in 2 60-min sessions. In each of these sessions, there were 60 sucrose deliveries delivered on a variable-time 60-s schedule. Next, a single lever press training session was given, in which each lever press was followed by sucrose delivery; each rat was removed from the chamber after approximately 50 presses. Then, to establish strong operant baseline lever-press responding, the rats received a single session in which lever presses were reinforced on a variable-interval 60-s schedule, followed by 4 sessions in which lever-pressing was reinforced on a variable-interval 120-s schedule. These and all subsequent sessions were 90 min in duration. No other stimuli were delivered.

Pavlovian fear conditioning began with two 90-min sessions designed to establish conditioned suppression to the target cue to be used in discrimination training and another cue to be used in a transfer test. Each session included one 1-min presentation of an intermittent (2 Hz) 1500-Hz tone and one 1-min presentation of a white noise, each reinforced with a 0.5-s, 0.5-mA shock. During the first 45 min of the next session the rats received 3 non-reinforced presentations of a 1-min illumination of the house light as a pretest of responding to that feature cue. Discrimination training began in the last 45 min of that session. The rats received a single 1-min tone presentation that is rewarded and 3 non-rewarded presentations of a serial compound consisting of a 1-min presentation of the house-light followed by the 1-min tone. During the remaining 47 discrimination training sessions, the rats received two rewarded tone presentations and six non-rewarded presentations of the light-tone compound. The trial sequences were randomized and changed daily; the inter-trial intervals averaged 11 min, ranging from 6 to 18 min. Finally, all rats received a non-reinforced probe test which examined responding to the tone and noise excitors and to serial compounds of those excitors with the light (2 presentations each of the tone, the noise, the light + tone compound and the light + noise compound). No shocks were delivered during this test regardless of stimulus identity. Because the light + noise trials were unique to the probe test, we present data only for the tone and light + tone trials. The measure of conditioning was a standard suppression ratio[42] computed by dividing the lever-press response rate during CS presentations by the sum of response rates during CS presentations and for 2 min prior to CS presentations. Discrimination performance was measured by constructing a difference score, suppression during the tone on light + tone compound trials minus suppression on tone-alone trials.

For both Experiments 1 and 2, we examined only performance in the reinforced and non-reinforced contexts at the end stages of training, as probe trials were not conducted over the entire course of training.

**Behavioral training: ferrets**. All experimental procedures conformed to standards specified and approved by the French Ministry of Research and the ethics committee for animal experimentation n°5. Adult female ferrets were housed in pairs in normal outside light cycle vivarium. After headpost implantation, ferrets were habituated to head-fixed holder for a week. They were then trained until they reached performance criterion. Two adult female ferrets were trained to discriminate 1.1 s-long click trains in different paradigms (one on low versus high rate click train discrimination and the other on regular versus irregular click train) in a Go/No-Go task under appetitive reinforcement. The first ferret was trained to discriminate between a high-frequency (24 Hz) foil stimulus and a low-frequency (4 Hz) target stimulus, with a response window of 1.85 s following the stimulus presentation, and performance was tracked in both contexts throughout learning. The second ferret was trained to discriminate between a 12 Hz irregular click-train (foil) and a 12 Hz regular click-train (target), with a response window of 0.8 sec following stimulus presentation. Performance on probe versus reinforced trials were only assessed at an early stage of training (trial 1–1150). Animals were rewarded with water (0.2 mL) for licking a waterspout in the response window. Licks during the foil response window were punished with a timeout, as well as

licks during the earlier part of the target click train (Early Window). We present and model only the learning trajectory of the first ferret, but we note that average behavioral performance was similar across animals.

All sounds were synthesized using a 100 kHz sampling rate, and presented through a free-field speaker that was equalized to achieve a flat gain. Clicks were mono-polar, rectangular pulses of 1 ms duration with amplitude set at 70 dB SPL. Behavior and stimulus presentation were controlled by custom software written in Matlab (MathWorks). Target and foil stimuli were preceded by an initial silence lasting 0.2 s (Ferret 1) and 0.5 s (Ferret 2) followed by the 1.1 s-long click trains. On each session, foil and target stimuli were randomly presented and kept constant through training.

**Reinforcement learning model.** We constructed a decision-making model that implements reinforcement-driven learning of stimulus-action associations[17], with a readout function that can be contextually modulated. The core model consisted of a sensory coding population which sends excitatory projections to a decision-making population through feed-forward inhibition (Fig. 3a). The sensory population consists of two tone-selective populations representing target (S+) and foil tones (S−), and one additional population that is tone-responsive but has no preference for targets or foils (S), consistent with the functional organization of auditory cortical networks[21,26–29]. The non-selective sensory population (S) captures any generalized stimulus-action associations, and greatly improves model performance. The three sensory populations projected to inhibitory (I) and excitatory populations (D) with plastic synapses in the decision-making area. The strengths of these synapses changed through reward-driven plasticity on a trial-by-trial basis; in reinforcement learning terms, the synaptic weights here represented the action values of the stimuli. The excitatory decision-making unit read out and compared the values of the two actions (Go and No-Go) by computing total synaptic input, i.e., the difference between direct excitation for a Go response, and feed-forward inhibition that promotes a No-Go response. This total input corresponded to a decision variable, which the decision-making unit transformed into an action through a noisy all-or-none activation function.

Sensory representations were assumed fixed during learning, and thus this layer formally reduces to a binary 3D vector $\mathbf{x} = [S\ S_+\ S_-]$ (i.e., the sensory representation layer can be represented as [1 0 1] during a foil trial). The instantaneous strengths of the projections from the sensory layer to the decision layer are determined by two 3D weight vectors, $\mathbf{W}_D = [w_{D/S}\ w_{D/S+}\ w_{D/S-}]$ and $\mathbf{W}_I = [w_{I/S}\ w_{I/S+}\ w_{I/S-}]$. We assumed that initial weights from the two tone –selective units were identical, but allowed the initial weights from the non-selective population to be independently determined. The inhibitory unit provides graded linear feed-forward inhibition to the decision unit. The decision unit reads out the utility values of the two possible actions, Go or No-go, by computing its net synaptic input. The probability of generating a Go decision is given by

$$P(y = 1|\mathbf{x}) = \frac{1}{1 + \exp(-(\mathbf{W}_D\mathbf{x}^T - \mathbf{W}_I\mathbf{x}^T)\sigma^{-1})} \quad (1)$$

where $\sigma$ is a parameter that regulates the stochasticity of behavioral decision making, analogous to the temperature parameter in canonical reinforcement models. We denote by $y$ the output of the decision unit, with $y = 1$ for a Go and $y = 0$ for a No-Go.

Synaptic weights from the sensory to the decision-making layer were updated at the end of each trial on the basis of the obtained reinforcement. Because of the relatively slower change in false-alarm rates than hit rates, we allowed the synaptic changes following rewarded and non-rewarded trials to have different learning rates $\alpha$ and $\alpha_{NR}$. To account for the learning delay observed in many individual animals, we followed Bathellier et al.[17] and utilized a multiplicative learning rule in which the learning rates are multiplied by synaptic strengths, so that strong synapses are updated more rapidly than weak synapses (Supplementary Fig. 11a). This multiplicative rule enabled the model to capture both exponential and sigmoidal learning trajectories, and predicts that the initial weights between the sensory representation layer and the decision circuitry regulates the general shape of learning trajectories for individual animals[17]. An additive model failed to account for the learning trajectories of individual animals (Supplementary Fig. 11b). Taken together, synaptic weights are strengthened and weakened according the following learning rules:

$$\text{Rewarded trials}: \begin{array}{l} \delta\vec{W}_{D,j} = \alpha\mathbf{W}_{D,j}(R - \kappa^{-1}(\mathbf{W}_D\mathbf{x}^T - \mathbf{W}_I\mathbf{x}^T))y \\ \delta\vec{W}_{I,j} = -\alpha\mathbf{W}_{I,j}(R - \kappa^{-1}(\mathbf{W}_D\mathbf{x}^T - \mathbf{W}_I\mathbf{x}^T))y \end{array} \quad (2)$$

$$\text{Unrewarded trials}: \begin{array}{l} \delta\vec{W}_{D,j} = \alpha_{NR}\mathbf{W}_{D,j}(R - \kappa^{-1}(\mathbf{W}_D\mathbf{x}^T - \mathbf{W}_I\mathbf{x}^T))y \\ \delta\vec{W}_{I,j} = -\alpha_{NR}\mathbf{W}_{I,j}(R - \kappa^{-1}(\mathbf{W}_D\mathbf{x}^T - \mathbf{W}_I\mathbf{x}^T))y \end{array} \quad (3)$$

where R represents the reward (−1 if not rewarded, 1 if rewarded), $\kappa$ is a parameter that regulates the asymptotic weights of each synapse, and $y$ is a Hebbian term that requires co-activation of pre-synaptic and post-synaptic terminals for synaptic modifications, as it does not provide any update if the decision neuron does not activate. During stochastic runs of the model, the target and foil stimuli were generated pseudorandomly with equal probability.

To account for the distinct learning trajectories in the reinforced context and the probe context, we extended the original model and introduced a 2D binary context vector $\vec{s}$, indicating whether the licktube was present [1 0] or absent [0 1]. For the inhibitory scaling model, feed-forward inhibition was scaled during reinforced trials during by the 2D vector $\vec{c}_I = [c_{I(\text{reinforced})}\ 1]$, such that:

$$P(y = 1|\vec{x}) = \frac{1}{1 + \exp(-(\mathbf{W}_D\mathbf{x}^T - (\mathbf{c}_I\mathbf{s}^T)(\mathbf{W}_I\mathbf{x}^T))\sigma^{-1})} \quad (4)$$

effectively shifting the decision-making unit's readout from its baseline state during the reinforced context. Other models tested were subject to similar context dependent switches applied as follows:

$$\text{Gain modulation}: P(y = 1|\vec{x}) = \frac{1}{1 + \exp(-(\mathbf{c}_G\mathbf{s}^T)(\mathbf{W}_D\mathbf{x}^T - \mathbf{W}_I\mathbf{x}^T)\sigma^{-1})} \quad (5)$$

$$\text{Threshold shift}: P(y = 1|\vec{x}) = \frac{1}{1 + \exp(-(\mathbf{W}_D\mathbf{x}^T - \mathbf{W}_I\mathbf{x}^T + \mathbf{c}_T\mathbf{s}^T)\sigma^{-1})} \quad (6)$$

$$\text{Excitatory scaling}: P(y = 1|\vec{x}) = \frac{1}{1 + \exp(-((\mathbf{c}_E\mathbf{s}^T)(\mathbf{W}_D\mathbf{x}^T) - \mathbf{W}_I\mathbf{x}^T)\sigma^{-1})} \quad (7)$$

Note that gain modulation is effectively equivalent to a modulation of the noise parameter. Throughout training, we probed the model after every 100 reinforced trials ($\vec{s} = [10]$) for its behavior across 100 probe ($\vec{s} = [01]$) trials. Because mice received no positive reinforcer during the probe context trials, we assumed that synaptic weights were not updated during these probe trials. This assumption allows probing to, theoretically, progress indefinitely to assess the baseline (non-scaled) behavior of the model, without altering the synaptic weights representing task knowledge. This allowed us to sample from the model during both behavioral contexts over the entire extent of learning.

**Modeling of rat behavioral data.** To generalize our model to the behavior of freely moving rats in a task without a binary choice point, we simply altered the readout function to yield a continuum of possible values for the percentage of time spent in the food cup. Rather than having the readout function yield the probability of a 'Go' response, we took this same value to indicate the percentage of time spent at the food cup. For example, when the original readout function yielded a probability of a 'Go' response as 65%, we converted this to mean 65% of time spent at the food-cup. The readout function in the rat behavioral task can thus be written as:

$$T = \frac{1}{1 + \exp(-(\mathbf{W}_D\mathbf{x}^T - \mathbf{W}_I\mathbf{x}^T)\sigma^{-1})} \quad (8)$$

where $T$ is the percentage of the trial spent in the food cup. This thus preserves most aspects of the original model (with the exception of being slightly less stochastic), including the readout function serving as a measure of the animal's bias toward one response given the stimulus.

**Model fitting.** All simulations and fitting procedures were performed in MATLAB. All tested models were fitted to data in both contexts simultaneously. To increase computational efficiency, we constructed a coarse-grained version of our model by assuming slow variations in the synaptic weights. During fitting, the model weights were updated in chunks of 10 trials, with stochasticity solely arising from the target and foil ratios in the given block. Trial ratios were pseudo-randomly drawn from a normal distribution ($\mu = 0.5$, $\sigma = 0.1$; <0.5 = F, >0.5 = T). During each trial block, the reinforced-context performance was calculated given the synaptic weights preceding the given block, and synaptic weights subsequently updated on the basis of the probability of false-alarm and hit trials during the given 10 trials. For example:

$$\delta W_{\frac{D}{S}+} = \alpha W_{\frac{D}{S}+}\left(R - \kappa^{-1}\left(W_{\frac{D}{S}+} + W_{\frac{D}{S}} - W_{\frac{I}{S}+} - W_{\frac{I}{S}}\right)\right)n_T P(y = 1|\mathbf{W}_D, \mathbf{W}_I) \quad (9)$$

where $n_T$ represents the number of target-tone trials in the given trial block, which is weighted by the probability of the model "licking" to the target tone given the current weights. The model was tested for the hit and false-alarm rates in the probe context. These approximations closely replicated the behavior of the fully stochastic model across a large number of runs, but required significantly less computational power. For each model, we minimized the Root Mean Square (RMS) error between the model performance and the behavioral S+ and S− response rates in both the reinforced and the probe context using Bayesian adaptive direct search (BADS[43]). BADS alternates between a series of fast, local Bayesian optimization steps and a systematic, slower exploration of a mesh grid.

To ensure a robust model fit to the acquisition and context-dependent expression of task knowledge, we excluded a small number of reinforced context training blocks during which a robust but temporary decline in satiety and/or motivation was observed. These were defined as training blocks during which false-alarm rates and hit rates both decreased by >30% with respect to the preceding and proceeding training blocks (2 training blocks total across 7 animals). Additionally, one probe training block was excluded during model fitting because an insufficient number of trials for robust analysis (10 trials total). All other probe training blocks consisted of at least 20 trials and were included in analysis. For every trial after the peak hit rate was reached in the probe context, we assumed each animal achieved perfect discrimination based on our evidence from three animals in which the

asymptotic hit rate was $92 \pm 4\%$ and the false-alarm rate was $3 \pm 3\%$. This assumption served a twofold function: firstly, it allowed the model to ignore the cessation of behavioral responses in the probe context; secondly, it effectively penalized the model for adopting a strategy in which it assumed that perfect expression of task knowledge could not be achieved in the probe context, despite continued training. To allow the model to center its average performance around the generalized learning trajectories, we applied a light lowpass filter to behavioral learning trajectories during fitting, with filter coefficients equal to 0.20 and 0.33 in the reinforced and probe context, respectively.

**Analysis of model results**. Decision variables were generated from the average synaptic weights of stochastic models on a trial-by-trial basis, and serve to highlight the effects of contextual factors. The trajectories of these variables illustrate the decision read-out function as training progresses, and are separated into target and foil trials. For example, the instantaneous value of each trajectory is thus defined as $\mathbf{W}_D\mathbf{x}^T - \mathbf{W}_I\mathbf{x}^T$ in the probe context. Error rates of each tested model were quantified as the sum of the RMS error between the model and behavioral learning trajectories across both behavioral contexts. For comparison, we ran stochastic models 200 times to capture the full extent of variance arising from random tone selection and noise in the decision read-out function.

To understand which of our parameter most strongly contributed to inter-individual variation observed in the reinforced context, we utilized a one-factor-at-a-time approach to examine how much each parameter could alter the learning curve versus how much real learning curves differed. First, we established the average parameters required to fit the average behavioral data (9 parameters). Next, we varied a single parameter (i.e., $c_I$) within the range corresponding to all of the individual animal fits (i.e., $c_I = 0.07$–$0.48$) and calculated the resulting error relative to the average fit for each value of the parameter (RMSE with respect to average behavior). We found the maximum error generated by this entire range of values (Maximum Model Error). We then calculated the maximum error within the behavioral data (Maximum Behavioral Error; RMSE of individual learning trajectories with respect to average learning trajectories), and defined explained variation as $\frac{\text{Maximum model error}}{\text{Maximum behavior error}}$. Finally, we performed this calculation for each of the model parameters ($\alpha$, $\alpha_{NR}$, $\sigma$, $\kappa$, $\mathbf{W}_E$, $\mathbf{W}_I$, $\mathbf{W}_{SE}$, $\mathbf{W}_{SI}$, and $c_I$). To determine how different parameters contributed to the model error, we divided our parameters into four groups: learning rates ($\alpha$, $\alpha_{NR}$), initial conditions ($\mathbf{W}_E$, $\mathbf{W}_I$, $\mathbf{W}_{SE}$, $\mathbf{W}_{SI}$), noise ($\sigma$), and inhibitory scaling ($c_I$). To remain conservative in our analysis, the parameter in each group that explained most variation was selected to be representative.

**Quantification and statistical analysis**. All statistical analyses were performed in MATLAB or GraphPad Prism 7. Data sets were tested for normality, and appropriate statistical tests applied as described in the text (for example, $t$-test for normally distributed data, Fischer's exact test for categorical observations, Mann Whitney $U$-test for non-parametric data, Friedman test with Dunn *post hoc* test for non-parametric data with repeated measurements, Geisser-Greenhouse correction for violations of sphericity). All statistical tests used were two-tailed. Model-variance designed to reflect the stochasticity of behavioral decision making was drawn from a standard normal distribution, and all model comparisons thus assumed normality. Shaded regions surrounding behavioral line-plots indicate $\pm$ s.e.m. unless otherwise stated. Shaded regions surrounding model line-plots indicate $\pm$ st.d. unless otherwise stated. Experimenters were not blind to the conditions of the experiments during data collection and analysis.

**Reporting summary**. Further information on research design is available in the Nature Research Reporting Summary linked to this article.

## Data availability

All behavioral data that underlies the findings of this study, as well as all code related to the modeling work, is available at: http://froemkelab.med.nyu.edu/sites/default/files/Data_NatComms.zip.

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

## Acknowledgements

We thank C. Firestone, C. Honey, J. Chen, K. Katlowitz, K. Narasimhan, and W.J. Ma for comments on earlier versions of this manuscript; W.J. Ma, R. Kiani, M. Long, and K. Louie for assistance with the conceptual model; J. Multani and J. Garmon for assistance with behavioral experiments. This work was funded by grants from NIDCD (DC009635 and DC012557), a Hirschl/Weill-Caulier Career Award, and a Howard Hughes Medical Institute Faculty Scholarship (R.C.F.); the NIDCD (DC05014) (K.V.K.); and the Programme Emergences of City of Paris, ANR grants ANR-17-ERC2-0005, ANR-16-CE37-0016, and the program "Investissements d'Avenir" ANR-10-LABX-0087 IEC and ANR-11-IDEX-0001-02 PSL Research University (S.O.); and the NIH training program in computational neuroscience (R90DA043849) (T.A.H.S.).

## Author contributions

K.V.K., Y.B., P.C.H., R.K., K.F., S.E. and E.S.P. performed behavioral experiments. T.A.H.S., S.O. and K.V.K. designed and T.A.H.S. implemented the theoretical model. K.V.K. and T.A.H.S. performed analysis. All authors discussed experiments and T.A.H.S, K.V.K., R.C. and S.O. wrote the manuscript.

## Additional information

**Competing interests:** The authors declare no competing interests.

