## [Peer Review File · Nature Communications]

Editorial Note: This manuscript has been previously reviewed at another journal that is not operating a transparent peer review scheme. This document only contains reviewer comments and rebuttal letters for versions considered at Nature Communications .

Reviewers' Comments:

Reviewer #2:

Remarks to the Author:

This is a revision of a manuscript that has been transferred from [REDACTED]
I was already convinced by the previous version, but this one got even better. In particular, I found the conceptual changes to the discussion section well done, and I enjoyed the clarifications made in the results section.

I only have minor comments:

- Fig. 2, legend: Please move the explanation of "Discrimination" from d) to b)
- For Figure 2 and associated text, I would still recommend to remove some of the specialist language, which I think is not necessary for the current purpose. E.g., the description of the various tasks reads very technical: "feature-negative discrimination task", "reverses its predictive quality such that no food is delivered", "In this feature-negative discrimination task, rats were first trained to press a lever for sucrose reinforcement, and Pavlovian fear conditioning procedures were subsequently superimposed on this operant lever pressing baseline."
- l. 551: sentence needs fixing: "The decision unit's (D) activity determines whether the probability that the model will respond."
- Fig. 4f: blank needed between Variation and (%)
- l. 590: In the legend, "Each parameter is constrained by the values given by individual animal fits" is not helpful because the sentence is unclear unless the reader has gone through the explanation in l. 338. I would recommend removing or clarifying the sentence.

Reviewer #3:

Remarks to the Author:

In the current revision of the paper the authors have made quite extensive highlighted changes to the text that seem very good to me and have added an additional experiment using lever presses to dissociate the learned action from the appetitive response. Briefly, I found this experiment to be well designed, persuasive and interesting complement to a strong paper with important insights into the assessment of learning. I agree that the new experiment provides important support for the authors' interpretation of the data: we can think of the probe trial block as a context that effectively dissociates variables determining the performance of the task from underlying associative knowledge (or to use the author's terminology and Tolman's terminology "latent" knowledge/learning). I believe the paper should be published in its current form with no additional changes absolutely required. Below I make some comments that the authors may or may not feel improves their manuscript and they might incorporate if they wish.

I was interested to understand a bit more about the probe data in Supp Figure 1. Whereas in the previous version of the task the retraction of the lick port was potentially recognized as a change context, in this version the context must be experienced (a hit lever press does not cause the retracted lick port to extend forward) as I understand the task. This suggests that the reduction in the action rate emerges over a few trials? Perhaps helping to explain why the action rate is higher in probe context in Sup Fig 1 than Fig 1. Also, this seems to be apparent in the ferret behavior in Supp Fig 3. In this case I was interested whether the term c_i in the model was allowed to evolve as a function of trials (i.e. with a learning rate) whether this might account for data better. It also might explain the somewhat smaller contribution of context in the model results in Supp Fig 3e. I thought some discussion of this could be valuable. I think it is still reasonable to conceptualize this as context,

but it is a bit distinct from how context is invoked in other learning paradigms where it is a context in which learning occurs, and generally not something with learned associations to action itself or requiring learning to recognize as a new context.

I think generally speaking the point the authors make that it is problematic to capture variability of acquisition of a task in a single learning parameter or initialization conditions alone is a very important one and this manuscript has perhaps the best demonstration of that limitation to date. In the much improved discussion the authors also discuss how to reconcile this with reinforcement learning and the often synonymous treatment of that with dopamine. On line 383 the authors begin an interesting discussion of implications for future circuit experiments. Although they understandably only briefly touch on dopamine there have been a number of recent papers in that literature that have also begun to show complex properties during acquisition of tasks consistent with multiple components of what behaviorally may look like a single trajectory. Three recent examples are 10.7554/eLife.21886, 10.7554/eLife.13665, and 10.1038/s41593-018-0245-7. In some of this work (including a recent preprint: 10.1101/520965) the authors used sensory preconditioning (a classic demonstration of latent learning and thus some connection to the arguments in this study) to reveal that dopamine supports only part of the knowledge or execution acquired during learning. There is perhaps more to discuss about how the context design exploited here might be used to further reveal aspects of latent knowledge or distinctions between latent knowledge and execution that have not been readily apparent or directly addressed previously in that literature - perhaps dissociating the roles of sensory areas that the authors are most interested in with other functions in "decision making" areas to which they refer.

REVIEWERS' COMMENTS:

Reviewer #2:

- Fig. 2, legend: Please move the explanation of "Discrimination" from d) to b)

- Fixed

- For Figure 2 and associated text, I would still recommend to remove some of the specialist language, which I think is not necessary for the current purpose. E.g., the description of the various tasks reads very technical: "feature-negative discrimination task", "reverses its predictive quality such that no food is delivered", "In this feature-negative discrimination task, rats were first trained to press a lever for sucrose reinforcement, and Pavlovian fear conditioning procedures were subsequently superimposed on this operant lever pressing baseline."

- We have removed some of this jargon to hopefully make the descriptions of the task clearer.

- l. 551: sentence needs fixing: "The decision unit's (D) activity determines whether the probability that the model will respond."

- Fixed.

- Fig. 4f: blank needed between Variation and (%)

- Fixed.

- l. 590: In the legend, "Each parameter is constrained by the values given by individual animal fits" is not helpful because the sentence is unclear unless the reader has gone through the explanation in l. 338. I would recommend removing or clarifying the sentence.

- We have removed this sentence from the manuscript.

Reviewer #3:

In the current revision of the paper the authors have made quite extensive highlighted changes to the text that seem very good to me and have added an additional experiment using lever presses to dissociate the learned action from the appetitive response. Briefly, I found this experiment to be well designed, persuasive and interesting complement to a strong paper with important insights into the assessment of learning. I agree that the new experiment provides important support for the authors' interpretation of the data: we can think of the probe trial block as a context that effectively dissociates variables determining the performance of the task from underlying associative knowledge (or to use the author's terminology and Tolman's terminology "latent" knowledge/learning). I believe the paper should be published in its current form with no additional changes absolutely required. Below I make some comments that the authors may or may not feel improves their manuscript and they might incorporate if they wish.

I was interested to understand a bit more about the probe data in Supp Figure 1. Whereas in the previous version of the task the retraction of the lick port was potentially recognized as a change context, in this version the context must be experienced (a hit lever press does not cause the retracted lick port to extend forward) as I understand the task. This suggests that the reduction in the action rate emerges over a few trials? Perhaps helping to explain why the action rate is higher in probe context in Sup Fig 1 than Fig 1. Also, this seems to be apparent in the ferret behavior in Supp Fig 3. In this case I was interested whether the term c_i in the model was allowed to evolve as a function of trials (i.e. with a learning rate) whether this might account for data better. It also might explain the somewhat smaller contribution of context in the model results in Supp Fig 3e. I thought some discussion of this could be valuable. I think it is still reasonable to conceptualize this as context, but it is a bit distinct from how context is invoked in other learning paradigms where it is a context in which learning occurs, and generally not something with learned associations to action itself or requiring learning to recognize as a new context.

- Thanks for these ideas. C_i in our implementation was fit as a scalar constant throughout learning since we are considering the impact of reinforcement to be constant throughout learning. C_i was fit as a constant to remain parsimonious in our analysis. If C_i is allowed to evolve over training the fits improve strongly, but we risk overfitting. In the future, and particularly as we collect neural data to test and further constrain the model, it will be interesting to consider how such a contextual parameter may evolve as a function of trials/learning.
- Regarding the definition of this as a context, we aim to be clear that we are referring to this as a reinforcement context. We agree that this is different than how context is typically invoked but appreciate that you agree that it is reasonable to conceptualize it as such.

I think generally speaking the point the authors make that it is problematic to capture variability of acquisition of a task in a single learning parameter or initialization conditions alone is a very important one and this manuscript has perhaps the best demonstration of that limitation to date. In the much improved discussion the authors also discuss how to reconcile this with reinforcement learning and the often synonymous treatment of that with dopamine. On line 383 the authors begin an interesting discussion of implications for future circuit experiments. Although they understandably only briefly touch on dopamine there have been a number of recent papers in that literature that have also begun to show complex properties during acquisition of tasks consistent with multiple components of what behaviorally may look like a single trajectory. Three recent examples are 10.7554/eLife.21886, 10.7554/eLife.13665, and 10.1038/s41593-018-0245-7. In some of this work (including a recent preprint: 10.1101/520965) the authors used sensory preconditioning (a classic demonstration of latent learning and thus some connection to the arguments in this study) to reveal that dopamine

supports only part of the knowledge or execution acquired during learning. There is perhaps more to discuss about how the context design exploited here might be used to further reveal aspects of latent knowledge or distinctions between latent knowledge and execution that have not been readily apparent or directly addressed previously in that literature - perhaps dissociating the roles of sensory areas that the authors are most interested in with other functions in “decision making” areas to which they refer.

- We thank the reviewer for the comments about variability and about the quality of the discussion. We intend to address these types of questions regarding dopamine through future experiments (and a potential review on this topic). The papers you highlighted are indeed very interesting and warrant a more in-depth incorporation into the broader dopamine literature.